# Radiofrequency Irradiation Attenuates High-Mobility Group Box 1 and Toll-like Receptor Activation in Ultraviolet B–Induced Skin Inflammation

**DOI:** 10.3390/molecules26051297

**Published:** 2021-02-28

**Authors:** Hyoung Moon Kim, Seyeon Oh, Jung Hyun Yoon, Donghwan Kang, Myeongjoo Son, Kyunghee Byun

**Affiliations:** 1Maylin Clinic, Goyang-si, Gyeonggi-do 10391, Korea; md.mac12@gmail.com; 2Department of Anatomy & Cell Biology, Gachon University College of Medicine, Incheon 21936, Korea; 3Functional Cellular Networks Laboratory, Lee Gil Ya Cancer and Diabetes Institute, College of Medicine, Gachon University, Incheon 21999, Korea; seyeon8965@gmail.com; 4Yonseifams Clinic, Seoul 03396, Korea; urohyun@hanmail.net; 5Jeisys Medical Inc., Seoul 08501, Korea; kang@jeisys.com

**Keywords:** skin inflammation, TLR, HMGB1, micro needling radiofrequency, PIH

## Abstract

Ultraviolet B (UVB) exposure activates various inflammatory molecules of keratinocytes in the epidermis layer. Such UVB-mediated skin inflammation leaves post-inflammatory hyperpigmentation (PIH). Reports show a close relationship between PIH and high-mobility group box 1 (HMGB1) and its receptors. General clinical treatments of PIH, such as oral medication and laser treatment, have reported side effects. Recent studies reported the effects of radiofrequency (RF) irradiation on restoring dermal collagen, modulating the dermal vasculature, and thickening the basement membrane. To validate how RF regulates the inflammatory molecules from UVB-irradiated keratinocytes, we used UVB-radiated keratinocytes and macrophages, as well as animal skin. In addition, we examined two cases of RF-irradiated skin inflammatory diseases. We validated the effects of RF irradiation on keratinocytes by measuring expression levels of HMGB1, Toll-like receptors (TLRs), and other inflammatory factors. The results show that the RF modulates UVB-radiated keratinocytes to secrete fewer inflammatory factors and also modulates the expression of macrophages from HMGB1, TLRs, and inflammatory factors. RF irradiation could alleviate inflammatory skin diseases in patients. RF irradiation can regulate the macrophage indirectly through modulating the keratinocyte and inflammatory molecules of macrophages reduced in vitro and in vivo. Although the study is limited by the low number of cases, it demonstrates that RF irradiation can regulate skin inflammation in patients.

## 1. Introduction

Skin absorption of ultraviolet B (UVB; 290–320 nm) can lead to skin inflammation [1], prompt DNA mutations [2], suppress the immune system [3], and cause melanogenesis [4]. When skin is exposed to UVB, melanocytes located in the stratum basale of the epidermis layer produce melanin in melanosomes, which is transferred from melanocytes to keratinocytes [4]. In another reaction, inflammatory processes are initiated through the secretion and production of inflammatory mediators, such as chemokines, cytokines, and prostaglandins in keratinocytes. Furthermore, macrophages and neutrophils infiltrate the damaged site [5,6].

In the UVB-exposed epidermis, epidermal growth factor receptor is increased to suppress apoptosis and to lead keratinocyte proliferation, which in turn leads to epidermal hyperplasia associated with increased cyclin D expression [7]. Interestingly, skin stress due to UVB exposure can trigger systemic neuroendocrine responses with direct hormonal, neural, and chemical consequences [8]. UVB exposure stimulates the secretion of β-endorphins, corticosterone, urocortin, and adrenocorticotropic hormones through the hypothalamic–pituitary–adrenal axis, all of which are related to immunosuppressive effects [9,10]. In contrast, the nuclear factor kappa-light-chain-enhancer of activated B cells (NF-κB), a prototypical pro-inflammatory regulator, is also activated and controls cell growth through activation of cyclin D1 expression [11].

UVB-radiated keratinocytes secrete various kinds of inflammatory molecules, including high-mobility group box 1 (HMGB1), interleukin (IL)-1α, IL-1β, IL-6, IL-8, and tumor necrosis factor-alpha (TNF-α) [12,13,14]. Under normal conditions, the HMGB1 is localized in the nucleus [15]. However, when the cell is damaged or exposed to inflammatory cytokines, HMGB1 migrates to the cytoplasm and is finally secreted from the cell. The process of HMGB1 synthesis and secretion has been characterized in UVB-exposed keratinocytes, and HMGB1 is important for the initiation and progression of UVB-induced skin inflammation [16]. As the released HMGB1 interacts with its receptors, such as Toll-like receptor (TLR) 2 or TLR4, or with advanced glycation end products of the macrophages or monocyte, the inflammatory process is accelerated by monocyte and macrophage recruitment and by the secretion of inflammatory factors, including IL-6, cyclooxygenase (COX)-2, and TNF-α through NF-κB expression [16,17]. In addition, lipopolysaccharide (LPS) induces TLR4/nucleotide-binding oligomerization domain (NOD)-like receptor pyrin domain-containing-3 (NLRP3) inflammasome signaling. The LPS and TLR4 binding activates NF-κB and leads to NLRP3 assembly and hyperactivation. Finally, the inflammasome complex activates caspase-1, pro-IL-1β, and pro-IL-18, providing a precursor to the development of the mature forms of these factors. These signals excessively increase the secretion of pro-inflammatory cytokines [17].

Among various skin inflammatory diseases, post-inflammatory hyperpigmentation (PIH) results either from the irregular dispersion of melanin (for example, pigmentary incontinence) or from melanin overproduction following skin inflammation [18]. Pigmentary incontinence results from the destruction of the basal epidermis layer [19], which allows the upper dermis to accumulate macrophages that phagocytize degenerating keratinocytes and melanocytes. Melanin secretion from these melanocytes is believed to be the cause of hyperpigmentation [20]. Various inflammatory cytokines, including IL-1α, IL-6, COX-2, and TNF-α, as well as TLRs and HMGB1, are associated with PIH [21].

Clinical approaches to PIH treatment include topical bleaching cream, oral medication, chemical peels, and laser treatment. But most of the treatment modalities have side effects, so other treatment options are needed. Radiofrequency (RF) irradiation with micro-invasive electrodes is effective for improving pigmentation disorders, thickening the basement membrane, providing vascular modulation, and increasing dermal collagen production [22,23]. One study has shown the effectiveness of RF irradiation for the treatment of skin inflammation, but only a few studies have explored the mechanism of RF’s effects on skin [24]. Therefore, in this study’s UVB-radiated mouse model, we examined whether RF irradiation can attenuate inflammatory signals by reducing the expression of HMGB1 and inflammatory factors in keratinocytes and by migrating macrophages in cultured cells. Finally, we also examined whether RF irradiation could clinically alleviate the inflammatory skin diseases of patients.

## 2. Results

### 2.1. Inhibitory Effect of RF on Inflammatory Cytokine Secreted by UVB-Effected Keratinocytes

The inflammatory cytokines are expressed by UVB radiation [25,26]. In this study, UVB-radiated keratinocytes were used to validate the inhibitory effects RF had on the expression of the inflammatory cytokines. The results showed the secretion expression levels of HMGB1, IL-6, IL-8, and prostaglandin E_2_ (PGE_2_) were higher in the UVB-radiated human primary keratinocytes (UVB group) than in normal controls (Con group) and were significantly decreased in UVB radiation of RF-irradiated cells (UVB/RF group; Figure 1A–D). These results are comparable to those from an animal model of UVB inflammation (Figure 1E–H).

### 2.2. Inhibition Effect on Macrophage Activation by RF-Irradiated Keratinocytes 

Keratinocytes radiated with UVB secrete inflammatory cytokines, which activate the macrophages in the dermis layer [26,27,28]. In this study, we evaluated whether the inflammatory cytokines secreted by UVB-radiated keratinocytes affect macrophage activation (Figure 2A). The results show the activation of macrophages decreased statistically with increased RF irradiation on keratinocytes (Figure 2B). Similar results were shown in an animal model of UVB radiation. The number of activated macrophages in the dermis layer of the UVB group was higher than that in the Con group and was decreased in the UVB/RF group by Iba1 staining (Figure 2C and Appendix A). The messenger RNA (mRNA) expression level of CD163 was higher in the UVB group than in the Con group and decreased significantly in the UVB/RF group (Figure 2D).

### 2.3. Inhibitory Effects on the TLR Pathway of Macrophages Affected by RF-Irradiated Keratinocytes 

When activated macrophages are affected by UVB-irradiated keratinocytes, they activate the TLR2 and TLR4 pathways and cause inflammatory skin diseases [29,30]. We confirmed the inhibitory effects of macrophages on TLRs by controlling UVB-radiated keratinocytes using RF. The TLR2 and TLR4 expression was significantly higher in macrophages affected by UVB-radiated keratinocytes than in those in the Con group, but the expression was significantly lower in macrophages affected by both UVB and RF (Figure 3A,B). Likewise, the mRNA expression levels of inflammatory factors NF-κB, TNF-α, IL-6, and interferon-gamma (IFN-γ) related to TLR2 and TLR4 also decreased significantly in the UVB/RF group compared to the UVB group (Figure 3C). We also validated the mRNA expression level in skin tissue in a UVB-radiated animal model (Figure 3D,E). The mRNA expression levels of TLR2 and TLR4 were higher in the UVB group than in the Con group and decreased in the UVB/RF group. Similar results were also seen in immunostaining (Appendix A). The skin mRNA and protein expression levels of inflammatory factors NF-kB, TNF-α, IL-6, and IFN-γ related to TLR2 and TLR4 also decreased significantly in the UVB/RF group compared to the UVB group (Figure 3F and Appendix A). 

### 2.4. The Inhibitory Effects of RF Irradiation on Macrophages Affect Keratinocyte Proliferation and Pigment Accumulation 

UVB causes melanin accumulation, induced by keratinocyte proliferation and skin inflammation [25,31]. In this study, we confirmed the keratinocyte proliferation is caused not only by UVB radiation directly but by the activated macrophages and RF irradiation that inhibit keratinocyte proliferation in the cell model. UVB-radiated macrophages and cell supernatants from the macrophages were used to treat keratinocytes for 24 h (Appendix A). The mRNA expression levels of cell cycle-related genes, such as p21, p53, and cyclin D1, increased in the UVB group and decreased in the UVB/RF group (Appendix A). The mRNA expression of Bcl2 as an anti-apoptotic regulator increased in the UVB group and decreased in the UVB/RF group (Appendix A). Furthermore, expression levels of the inflammatory factors inducible nitric oxide synthase (iNOS) and prostaglandin E_2_ (PGE_2_) were higher in the UVB group than in the Con group and were lower in the UVB/RF group (Appendix A). Based on the results in our animal model, we validated epidermal keratinocyte proliferation in the skin from inflammation induced by UVB. We confirmed that the number of PCNA-positive cells was highest in the UVB group and was lower in the UVB/RF group (Figure 4A). Similar results were seen in the skin tissue for the mRNA expression levels in cell cycle-related genes p21, p53, and cyclin D1. Likewise, Bcl2 and iNOS, PGE_2_ inflammatory factors secreted from keratinocytes, showed similar results to those seen in the cell (Figure 4B–D). We confirmed that the effects of RF inhibited melanin accumulation in the inflammation model (Figure 4E), and we saw the attenuating effect on melanin accumulation in patients (Figure 4F).

## 3. Discussion

Although UVB-induced PIH and atopic dermatitis animal models are not well established, UVB-radiated animal models are widely used for studying skin inflammatory diseases. In this study, UVB exposure increased keratinocyte proliferation and expression of keratinocyte-derived inflammatory cytokines. In one study, RF irradiation had anti-proliferative effects on human breast cancer cells by regulating the expression of the cell cycle-related proteins p53, p21, cyclin A, cyclin B1, and cyclin D1 [32]. RF irradiation significantly impairs G2/M arrests in the cell cycle progression. In line with these data, those results showed the increased expression of p21, p53, and cyclin D1 in UVB-radiated mouse skin and UVB-radiated keratinocytes and their reduced expressions in a UVB/RF-irradiated mouse model and cultured cells (Appendix A).

The anti-proliferative effects of RF irradiation can decrease inflammatory cytokine expression in skin inflammation. The results show that expression levels of the inflammatory cytokines IL-6, IL-8, and HMGB1 were significantly increased in UVB-radiated mouse skin and keratinocytes and significantly decreased when UVB was supplemented by RF irradiation (Figure 1). In addition, the expression levels of HMGB1 receptors, such as TLR2 and TLR4, were also increased in UVB-radiated mouse skin but were statistically decreased in UVB/RF-irradiated mouse skin (Figure 3). The results are comparable to previous findings that RF irradiation reduced TNF-α expression in an acne-induced rabbit ear model, but there are few studies showing that RF modulates the expression of HMGB1 and TLRs, which are key molecules in skin inflammation [33].

We designed an in vitro model to validate how RF irradiation attenuates the inflammatory process and activates macrophages through modulating keratinocytes. In the model, the supernatants of RF-irradiated, UVB-radiated keratinocytes were used to treat macrophages. After 24 h of this treatment, the inflammatory cytokine levels of macrophages and the expression levels of TLRs were decreased in the human monocyte cell line (THP-1). Interestingly, UVB-radiated macrophages secreted some factors which can induce keratinocyte proliferation, and proliferative keratinocytes expressed inflammatory factors, such as iNOS and PGE_2_ (Appendix A). Therefore, RF irradiation can both directly regulate macrophage activation and indirectly regulate macrophage activation by modulating keratinocytes (Figure 2 and Appendix A). The results show that RF irradiation not only alleviates secreting cytokines in UVB-radiated keratinocytes but also attenuates the inflammatory process caused by macrophages.

PIH can be treated medically or non-medically; the latter group of treatments includes chemical peels and laser treatment [34,35]. But PIH patients usually suffer from long-lasting symptoms, with treatments resulting in only slow improvement and with frequent treatment having side effects. Thus, a safe and effective therapeutic modality is needed. Usually, PIH is induced via sequential events. Basement membrane damage is caused by inflammation through various external or internal stimuli, then epidermal inflammatory changes develop, and melanocytes are displaced downward into the dermis so that dermal macrophages phagocytose melanin-laden melanocyte in the dermis. Treatment of PIH should include (1) avoiding causative factors, for example allergens and infection sources; (2) removing inflammatory factors; (3) decreasing melanin content; (4) repairing the damaged basement membrane; and (5) improving the dermal environment. As far as we know, RF treatment for PIH (1) reduces pigmentation; (2) thickens the basement membrane; (3) induces dermal collagen production; and (4) modulates the dermal vasculature [23,34]. Our results also show that RF irradiation for PIH treatment is very effective and appropriate to the pathophysiology of PIH (Figure 4). After laser treatment, many doctors encounter PIH side effects. Dark skin color and chronic dermatitis are especially high-risk factors for PIH [36]. Ablative, high-power laser treatment also has high risks for PIH. In most cases, PIH improves spontaneously, but epidermal PIH lasts for at least 6 months, so its treatment is very important.

Chronic dermatitis not only induces hyperpigmentation but also leads to long-term usage of antihistamine drugs, steroid ointments, or other drugs. RF irradiation treatment is advantageous because it improves hyperpigmentation and allows patients to discontinue the use of antihistamine and steroid drugs or ointments.

## 4. Materials and Methods

### 4.1. UVB-Induced Skin Inflammation Models

#### 4.1.1. Animal Model (In Vivo)

We maintained male, HRM-2, melanin-possessing hairless mice (6 weeks old) in a temperature-controlled room (24 °C, 50% humidity) under a 12/12-h light–dark cycle. For this research, we used 6 mice per group. This research was approved by Gachon University and executed in accordance with the guidelines of the Institutional Animal Care and Use Committee (IACUC; No. LCDI-2018-0094). To induce skin inflammation, mice were regularly exposed to UVB for 5 min every 2 days for 10 days and then for 5 min every day for the following 4 days. At 14 days after exposure began, the mice were RF-irradiated (2 MHz, 10 W for 100 ms), after which they were exposed to UVB every 2 days. The skin tissues were collected after 28 days of RF irradiation.

#### 4.1.2. Cell Model (In Vitro)

Human primary epidermal keratinocytes (HEKn; ATCC, Manassas, VA, USA) and a human monocyte cell line (THP-1; ATCC) were used. The HEKn were cultivated with a growth cell medium (ATCC) and a keratinocyte growth kit (ATCC). The THP-1 were cultivated with high-glucose Dulbecco’s Modified Eagle Medium (Welgene, Gyeongsan-si, Korea) with 10% fetal bovine serum (Millipore, Burlington, MA, USA) and 1% penicillin/streptomycin (Welgene). All cells were maintained at 37 °C under 5% CO_2_. For the in vitro model, the cells were exposed to UVB for 5 min, were RF irradiated (2 MHz, 10 W, 100 ms), and were cultured for 24 h.

### 4.2. RF Irradiation Conditions for Patients

In case 1, RF irradiation was performed 2 times with insulated microelectrodes, with a 2-week interval. The irradiation consisted of 2 MHz of a bipolar alternating-current RF with 16 needle-tipped, insulated electrodes at a depth of 0.75 mm with an on time of 50 ms and an off time of 30 ms at 5 W in consecutive pulses, with 10 shots to each side.

In case 2, RF irradiation was performed 10 times with micro invasive electrodes, with a 2-week interval. The irradiation consisted of 2 MHz of a bipolar alternating-current RF with 16 needle-tipped, insulated electrodes at a depth of 0.75 mm with an on time of 50 ms and an off time of 30 ms at 5 W in consecutive pulses, for about 400 shots to the full face and neck.

### 4.3. Devices for RF Irradiation

The microneedling system (Potenza, Jeisys medical Inc. Seoul, Korea) used for animal experiments and clinical treatment is a bipolar pulsed-type electrode array RF device that allows the user to select the number of pulses, 1 MHz or 2 MHz, monopolar or bipolar. The impedance matching system was applied to measure the compensation value by measuring the impedance “automatically,” and proper output was irradiated through 16 needle tips. Pulsed-type, bipolar, alternating current oscillations, emitting 2 MHz RF, were used in the animal experiment (type A) and clinical treatment (type B). Single pulse-type bipolar RF devices were used for the animal experiment (type A): 100 ms was used for the on-time pulse duration at a power density of 10 W/pulse. Dual pulse-type bipolar RF devices were used for clinical treatment (type B): 50 ms was used for the on-time pulse duration, with 30 ms for the off-time pulse duration, at a power density of 5 W/pulse. The duty cycle was 130 ms. We conducted the investigation and treatment with disposable tips (size: 10 mm × 10 mm) comprising 16 invasive micro needle electrodes that were designed to a length of 13.6 mm, diameter of 250 mm, and needle-to-needle distance of 1.3 mm. This tip type was approved by the North American Science Associates (NAMSA, Northwood, OH, USA) via a biological compatibility test.

### 4.4. Cell Proliferation Measurement

To measure the viability of HEKn cells, they were maintained on a 96-well multi-plate (10^4^ cells/well) overnight, then exposed to UVB for 5 min, subjected to RF irradiation overnight, and cultured for 24 h. The HEKn cells were washed with calcium and magnesium-free phosphate-buffered saline (PBS), and a water-soluble tetrazolium salt-1 solution (DoGenBio, Seoul, Korea) was added (100 μL/well) for 4 h while the cells were in a cell incubator. Absorbance was measured at 450 nm in a VERSAmax tunable microplate reader (Molecular Devices, San Jose, CA, USA).

### 4.5. Sample Preparation

#### 4.5.1. Paraffin-Embedded Tissue Section Processing

Mice skin tissues were washed with cold PBS and stored in 4% paraformaldehyde (Biosesang, Seongnam-si, Korea) at 4 °C for 12 h. The fixed tissue was washed for 12 h for embedding, and a paraffin block was processed using a tissue processor (Shandon; ThermoFisher Scientific, Waltham, MA, USA). The blocks were cut into sections 10 μm thick using a microtome (Leica, Wetzlar, Germany) and dried at 45 °C overnight. The paraffin blocks were passed through xylene and four concentrations of ethanol (100%, 95%, 80%, and 70%) to prepare them for staining.

#### 4.5.2. RNA Extraction and cDNA Synthesis

In skin tissue, total RNA was isolated using RNAiso Plus (Takara, Shiga, Japan) according to the manufacturer’s instructions. The extracted RNA pellets were finally washed with a high-quality wash of 70% ethanol, were dried for 5 min, and were dissolved in 30 μL of diethyl pyrocarbonate-treated water. The extracted RNA was quantified using a Nanodrop 2000 (ThermoFisher Scientific) and was synthesized using a cDNA synthesis kit (Takara) according to the manufacturer’s instructions.

### 4.6. Quantitative Real-Time Polymerase Chain Reaction

The synthesized cDNA was used for quantitative real-time polymerase chain reaction (qRT-PCR). The qRT-PCR mixtures contained 5 µL SYBR Green reagent (Takara), 1 µg cDNA template in 2 µL, and a 10 pmol primer in 0.8 µL and were dispensed in each well of a 384-well multi-plate and then analyzed using CFX386 Touch (Bio-Rad, Hercules, CA, USA). Validated genes are listed in Appendix A.

### 4.7. Immunostaining Using 3,3-Diaminobenzidine

To block endogenous peroxidase, skin tissue slides were incubated in 0.3% hydrogen peroxide at room temperature for 30 min. The slides were rinsed 3 times with PBS and then incubated in normal animal serum with antibodies overnight to prevent antibody non-specific binding. The slides were then incubated with a proliferating cell nuclear antigen (PCNA) antibody and an ionized calcium-binding adaptor molecule 1 (Iba1) antibody for 12 h and washed three times with PBS. Tissue slides were treated with a biotinylated secondary antibody using the ABC kit (Vector Laboratories Inc, Burlingame, CA, USA) for 2 h, then washed 3 times with PBS. The tissue slides were incubated with 3,3’-diaminobenzidine (Sigma–Aldrich, St. Louis, MO, USA) for 15 min and washed with running water. To stain nuclei, tissue slides were immersed in hematoxylin solution for 1 min, then mounted with a Dibutylphthalate polystyrene xylene (DPX) mounting solution (Sigma–Aldrich). The stained tissues were photographed under an optical microscope (Olympus Optical Co., Korea) and analyzed with the Image J (NIH, Bethesda, MD, USA) software.

### 4.8. Statistical Analysis

SPSS version 22 (IBM Corporation, Armonk, NY, USA) was used to calculate significant differences between groups. Data were analyzed using a Kruskal–Wallis test (non-parametric statistics) and a Mann–Whitney *U* test for post hoc data; *p* values ≤ 0.05 were considered to indicate statistical significance.

## Figures and Tables

**Figure 1 molecules-26-01297-f001:**
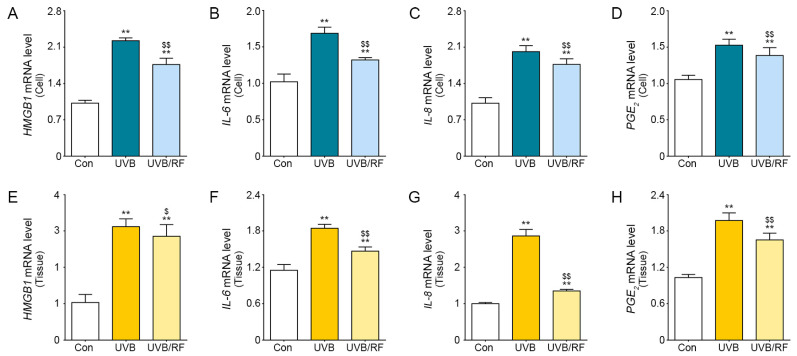
The inhibitory effects of radiofrequency (RF) irradiation on the expression of inflammatory cytokines from ultraviolet B (UVB)-radiated keratinocytes. The graphs show mRNA levels of inflammatory-related genes, including high-mobility group box 1 (HMGB1), interleukin (IL)-6, IL-8, and prostaglandin E_2_ (PGE_2_), in UVB-radiated (**A**–**D**) human primary epidermal keratinocytes (HEKn) and (**E**–**H**) mouse skin. All levels were confirmed by qRT-PCR. ** *p* < 0.01 vs. Con group; $ *p* < 0.05 and $$ *p* < 0.01 vs. UVB group. Results are presented as means ± standard deviations. Con, sham control; HEKn, human primary epidermal keratinocytes; HMGB1, high-mobility group box 1; IL, interleukin; mRNA, messenger RNA; PGE_2_, prostaglandin E_2_; qRT-PCR, quantitative real-time polymerase chain reaction; RF, radiofrequency; UVB, ultraviolet B.

**Figure 2 molecules-26-01297-f002:**
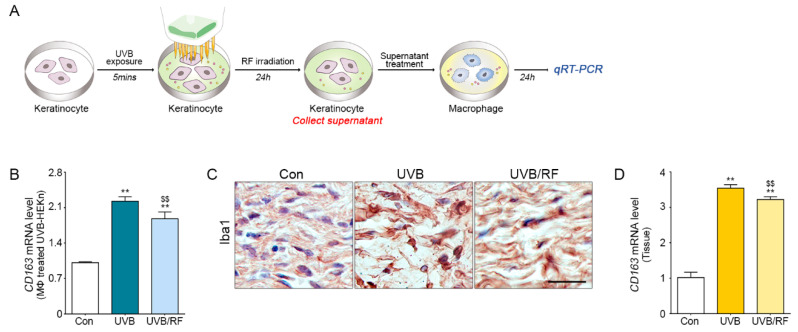
Inhibitory effects of RF irradiation on macrophage activation through keratinocyte modulation. (**A**) Diagram of an in vitro model to examine the effects of micro needling RF irradiation to UVB-radiated keratinocytes on the inflammatory response in macrophages. After exposing keratinocytes to UVB or UVB/RF, the cell culture supernatants were used to treat macrophages. (**B**) The mRNA levels of CD163, a dermal macrophage marker, were measured in macrophage-affected UVB-radiated keratinocytes. (**C**) The level of Iba1 protein, a macrophage activation marker, was detected in the skin dermis of UVB-radiated mice. (**D**) The mRNA levels of CD163 were detected in the skin of a UVB-radiated mouse. Scale bar = 100 µm. Magnification, ×40. ** *p* < 0.01 vs. Con group; $$ *p* < 0.01 vs. UVB group. Results are presented as means ± standard deviations. Con, sham control; mRNA, messenger RNA; qRT-PCR, quantitative real-time polymerase chain reaction; RF, radiofrequency; UVB, ultraviolet B.

**Figure 3 molecules-26-01297-f003:**
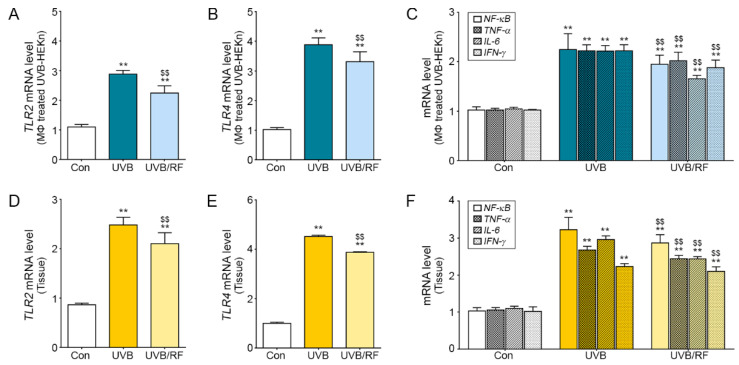
Inhibitory effects of RF irradiation on Toll-like receptor 2 (TLR2) and TLR4 expression in activated macrophages through keratinocyte modulation. The graphs show mRNA levels of TLR2, TLR4, and inflammation-related factors, including nuclear factor kappa-light-chain-enhancer of activated B cells (NF-κB), necrosis factor-alpha (TNF-α), IL-6, and interferon-gamma (IFN-γ), measured in (**A**–**C**) macrophages affected by UVB-radiated keratinocytes and (**D**–**F**) the skin dermis of a UVB-radiated mouse. ** *p* < 0.01 vs. Con group; $$ *p* < 0.01 vs. UVB group. Results are presented as means ± standard deviations. Con, sham control; IL-6, interleukin 6; IFN-γ, interferon-gamma; mRNA, messenger RNA; NF-κB, nuclear factor kappa-light-chain-enhancer of activated B cells; RF, radiofrequency; TLR, Toll-like receptor; TNF-α, tumor necrosis factor-alpha; UVB, ultraviolet B.

**Figure 4 molecules-26-01297-f004:**
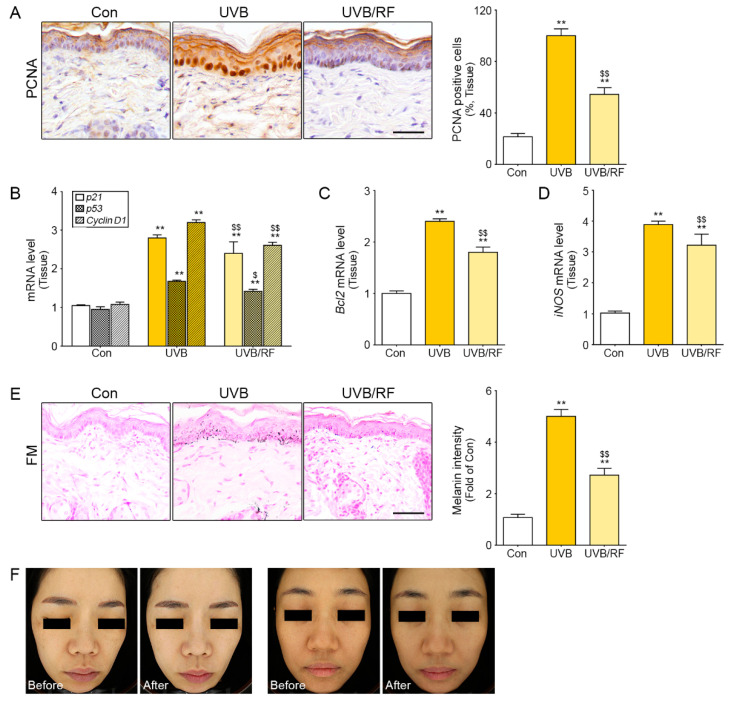
RF irradiation-induced macrophage activation, which had inhibitory effects on keratinocyte proliferation and pigmentation. (**A**) Keratinocyte proliferation in the epidermis of mouse skin was confirmed by detecting a proliferating cell marker (PCNA). The number of PCNA-positive cells was calculated from representative images. Scale bar = 100 µm. Magnification, ×40. (**B**–**D**) The graphs show mRNA levels of cell cycle-related genes (p21, p53, and cyclin D1), an apoptosis regulator gene (Bcl2), and pro-inflammatory factors in the skin dermis of a UVB-radiated mouse. (**E**) The melanin amount was confirmed in the epidermis by Fontana Masson staining and by quantifying the melanin amounts from representative images. Scale bar = 100 µm. Magnification, ×40. (**F**) In case 1, on the left, post-laser toning induced the development of post-inflammatory hyperpigmentation (PIH); melasma with background hyperpigmentation can be seen on the face. After RF irradiation, PIH disappeared, although melasma remained. In case 2, on the right, chronic dermatitis with hyperpigmentation in the forehead, periorbital, perioral, and neck areas was seen before RF irradiation. After RF irradiation, the hyperpigmentation decreased. ** *p* < 0.01 vs. Con group; $ *p* < 0.05 and $$ *p* < 0.01 vs. UVB group. Results are presented as means ± standard deviations. Con, sham control; mRNA, messenger RNA; PCNA, proliferating cell nuclear antigen; PIH, post-inflammatory hyperpigmentation; RF, radiofrequency; UVB, ultraviolet B.

## Data Availability

All data is contained within the article.

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
