# Peer review of "Radiofrequency Irradiation Attenuates High-Mobility Group Box 1 and Toll-like Receptor Activation in Ultraviolet B–Induced Skin Inflammation"

_molecules, 2021, doi:10.3390/molecules26051297_

Round 1
Reviewer 1 Report
In this study the effect of RF irradiation in attenuating the inflammatory response as well as the reduction of HMGB1 expression was investigated in murine machrophages, in a mouse models and in two patients.
The manuscript is well structured and the study is potentially interesting. However, some parts need clarifications
- page 2 lane 65. The melanin secretion from melanocytes is the cause of hyperpigmentation and not the consequence, as the author stated.
- page 2 lane 76. The authors should specify that the reduction in HMGB1 expression and inflammatory factors were observed in keratinocytes.
- Introduction. As the title suggests, the TLRs activation is implicated in skin inflammation. Please include a short description of TLR involvement in inflammatory response (Molecules 2019 Dec 10;24(24):4523. doi: 10.3390/molecules24244523)
- page 2, lane 83. The reference number should be included in square brackets.
- page 2 lane 86. Please specify that human primary keratinocytes were UVB radiated.
- pag. 3. Legend to the figure 1. Please correct PDE2 with PGE2. Are keratinocytes human epidermal?
- page 3 lane 101. The sentence should be …….”we validate whether the inflammatory…”.
- In this study the authors examined the mRNA expression of proinflammatory mediators. The quantification of cytokines should be performed (for ex by ELISA test) and protein levels of NFkB should be analyzed. The gene expression alone doesn’t explain the activation of intracellular signaling pathways. Again, Iba1 is an important marker of inflammation. Some pictures with immunostaining for Iba 1 should be included.
- page 4 lane 156. Please delete the sentence probably added for a mistake.
Although a more elevated number of patients should be included the study provides an interesting research topic for the potential importance to clinical practice.
Author Response
Response to Reviewer 1 Comments
In this study the effect of RF irradiation in attenuating the inflammatory response as well as the reduction of HMGB1 expression was investigated in murine machrophages, in a mouse models and in two patients.
The manuscript is well structured and the study is potentially interesting. However, some parts need clarifications.
Point 1: page 2 lane 65. The melanin secretion from melanocytes is the cause of hyperpigmentation and not the consequence, as the author stated.
Response 1: We appreciated with this comment. As your comment, ‘Melanin secretion from these melanocytes is believed as consequences of hyperpigmentation’ changed to ‘Melanin secretion from these melanocytes is believed to be the cause of hyperpigmentation’ in introduction section lane 77.
|
Introduction section, Page 2, Line 76 in revised manuscript [Previous version] Melanin secretion from these melanocytes is believed as consequences of hyperpigmentation [15].
[New version] Melanin secretion from these melanocytes is believed to be the cause of hyperpigmentation [20]. |
Point 2: page 2 lane 76. The authors should specify that the reduction in HMGB1 expression and inflammatory factors were observed in keratinocytes.
Response 2: We appreciated with this comment. As your comment, ‘Therefore, in this study, we examined whether RF irradiation can attenuate inflammatory signal by reducing the expression of HMGB1 and inflammatory factors and macrophage migration in in cultured cells and a UVB-radiated mouse model.’ changed to ‘Therefore, in this study’s UVB-radiated mouse model, we examined whether RF irra-diation can attenuate inflammatory signals by reducing the expression of HMGB1 and inflammatory factors in keratinocytes and by migrating macrophages in cultured cells.’ in introduction section lane 87.
|
Introduction section, Page 2, Line 87 in revised manuscript [Previous version] Therefore, in this study, we examined whether RF irradiation can attenuate inflammatory signal by reducing the expression of HMGB1 and inflammatory factors and macrophage migration in in cultured cells and a UVB-radiated mouse model. [New version] Therefore, in this study’s UVB-radiated mouse model, we examined whether RF irra-diation can attenuate inflammatory signals by reducing the expression of HMGB1 and inflammatory factors in keratinocytes and by migrating macrophages in cultured cells. |
Point 3: Introduction. As the title suggests, the TLRs activation is implicated in skin inflammation. Please include a short description of TLR involvement in inflammatory response (Molecules 2019 Dec 10;24(24):4523. doi: 10.3390/molecules24244523).
Response 3: We appreciated with kind comment. As you pointed out, we checked the reference paper you recommended and added the content in Introduction section.
|
Introduction section, Page 2, Line 60 in revised manuscript [Previous version] As the released HMGB1 interacts with its receptors such as receptor of Toll-like receptor (TLR) 2, or TLR4 and advanced glycation end products (RAGE) of the macrophages or monocyte, the inflammatory process is accelerated by monocyte and macrophage re-cruitment and secreting inflammatory cytokines including IL‐6 and TNF-α through NF-κB expression [16]. [New version] As the released HMGB1 interacts with its receptors, such as Toll-like receptor (TLR) 2 or TLR4, or with advanced glycation end products of the macrophages or monocyte, the inflammatory process is accelerated by monocyte and macrophage recruitment and by the secretion of inflammatory factors, including IL‐6, cyclooxygenase (COX)-2, and TNF-α through NF-κB expression [16,17]. In addition, lipopolysaccharide (LPS) induces TLR4/nucleotide-binding oligomerization domain (NOD)-like receptor pyrin domain-containing-3 (NLRP3) inflammasome signaling. The LPS and TLR4 binding activates NF-κB and leads to NLRP3 assembly and hyperactivation. Finally, the in-flammasome complex activates caspase-1, pro-IL-1β, and pro-IL-18, providing a pre-cursor to the development of the mature forms of these factors. These signals exces-sively increase the secretion of pro-inflammatory cytokines [17]. |
|
|
|
Reference section, Page 10, Line 388 in revised manuscript 17. Taticchi, A.; Urbani, S.; Albi, E. et al. In Vitro Anti-Inflammatory Effects of Phenolic Compounds from Moraiolo Virgin Olive Oil (MVOO) in Brain Cells via Regulating the TLR4/NLRP3 Axis. Molecules 2019, 24, 4523. |
Point 4: page 2, lane 83. The reference number should be included in square brackets.
Response 4: We appreciated with this comment. As your comment, reference number was included in square brackets.
|
Introduction section, Page 3, Line 95 in revised manuscript [Previous version] The inflammatory cytokines are expressed by UVB radiated17. [New version] The inflammatory cytokines are expressed by UVB radiatedation [25, 26]. |
Point 5: page 2 lane 86. Please specify that human primary keratinocytes were UVB radiated.
Response 5: We appreciated with this comment. As your comment, the human primary keratinocytes were specified in manuscript.
|
Results section, Page 2, Line 97 in revised manuscript [Previous version] The results show the expression level of HMGB1, IL-6, IL-8 and prostaglandin E2 (PGE2) secreted by UVB radiated keratinocyte (UVB) was higher than normal control group (Con) and significantly decreased in UVB radiation with RF irradiated cell group (UVB/RF) group (Figure 1A-D). [New version] The results show the secretion expression levels of HMGB1, IL-6, IL-8, and prostaglan-din E2 (PGE2) were higher in the UVB-radiated human primary keratinocytes (UVB group) than in normal controls (Con group) and were significantly decreased in UVB radiation of RF-irradiated cells (UVB/RF group; Figure 1A-D). |
Point 6: pag. 3. Legend to the figure 1. Please correct PDE2 with PGE2. Are keratinocytes human epidermal?
Response 6: We appreciated with this comment. As your comment, PDE2 was corrected with PGE2 and keratinocytes are human epidermal origin (cell datasheet, see below). This information was added in Figure 1 legend.
|
Figure 1 legend, Page 3, Line 105 in revised manuscript [Previous version] (A-D) The graphs showing mRNA levels of inflammatory related genes including HMGB1, IL-6, IL-8 and PDE2 in UVB-radiated human primary keratinocytes (HEKn) and these were confirmed by qRT-PCR. [New version] (A-D) The graphs showing mRNA levels of inflammatory related genes including HMGB1, IL-6, IL-8 and PGE2 in UVB-radiated human primary epidermal keratinocytes (HEKn) and these were confirmed by qRT-PCR. |
|
|
|
ATCC cell datasheet
|
Point 7: page 3 lane 101. The sentence should be …….”we validate whether the inflammatory…”.
Response 7: We appreciated with this comment. As your comment, we validate whether the inflammatory…” changed to “we validated whether the inflammatory…” in introduction section.
|
Results section, Page 3, Line 113 in revised manuscript [Previous version] In this study, we validated the inflammatory cytokines secreted by UVB radiated keratinocytes affect macrophage activation (Figure 2A). [New version] In this study, we evaluated whether the inflammatory cytokines secreted by UVB-radiated keratinocytes affect macrophage activation (Figure 2A). |
Point 8: In this study the authors examined the mRNA expression of proinflammatory mediators. The quantification of cytokines should be performed (for ex by ELISA test) and protein levels of NFkB should be analyzed. The gene expression alone doesn’t explain the activation of intracellular signaling pathways. Again, Iba1 is an important marker of inflammation. Some pictures with immunostaining for Iba 1 should be included.
Response 8: We appreciated and agree with this comment. As you have pointed out, to accurately identify intracellular signalling and activation of proinflammatory mediators, you should identify proteins rather than RNA. Therefore, we performed immunohistochemistry to accurately confirm the expression position of the protein in the mouse skin tissue, and the results can be confirmed below (supporting figure 1). Similar to the mRNA results, the expression levels of NF-kB, TNF-a, IL-6, and IFN-g proteins significantly increased after UVB treatment, but decreased statistically after RF treatment. We added this result to supplementary figure 4 and Results section.
|
Supplementary figure 4 and legend in revised supplementary manuscript
Figure S4. Inhibitory effects of RF irradiation on inflammatory mediator expression in vivo (A) Immunohistochemistry result showing NF-κB expression in the skin dermis of UVB-radiated mouse and graph showing intensity of NF-κB from representative results. (B) Immunohistochemistry result showing TNF-α expression in the skin dermis of UVB-radiated mouse and graph showing intensity of TNF-α from representative results. (C) Immunohistochemistry result showing IL-6 expression in the skin dermis of UVB-radiated mouse and graph showing intensity of IL-6 from representative results. (D) Immunohistochemistry result showing IFN-γ expression in the skin dermis of UVB-radiated mouse and graph showing intensity of IFN-γ from representative results. Scale bar =100 µm. Magnification, ×40. **, P < 0.01 and ***, P < 0.001 vs. Con group; $, P < 0.05 and $$, P < 0.01 vs. UVB group. Results are presented as means ± SD. Con, sham control; RF, radiofrequency; NF-κB, nuclear factor kappa-light-chain-enhancer of activated B cells; TNF-α, tumor necrosis factor alpha; IL-6, interleukin-6; IFN-γ, interferon-gamma; UVB, ultraviolet B |
|
|
|
Results section, Page 4, Line 135 in revised manuscript [Previous version] The mRNA expression level of inflammatory factor NF-kB, TNF-α, IL-6 and IFN-γ related with TLR2, 4 also decreased significantly in UVB/RF group than those of UVB group in skin (Figure 3F). [New version] The skin mRNA and protein expression levels of inflammatory factors NF-kB, TNF-α, IL-6, and IFN-γ related to TLR2 and TLR4 also decreased significantly in the UVB/RF group compared to the UVB group (Figure 3F and Figure S4). |
Point 9: page 4 lane 156. Please delete the sentence probably added for a mistake.
Response 9: We appreciated with scrupulous point. The sentence was deleted because it was entered incorrectly.
|
Results section, Page 4, Line 170 in revised manuscript [Previous version] This section may be divided by subheadings. It should provide a concise and precise description of the experimental results, their interpretation, as well as the experimental conclusions that can be drawn. [New version] This section may be divided by subheadings. It should provide a concise and precise description of the experimental results, their interpretation, as well as the experimental conclusions that can be drawn. |
Point 10: Although a more elevated number of patients should be included the study provides an interesting research topic for the potential importance to clinical practice.
Response 10: We appreciated with your comment. Through this study, we confirmed the effect of reducing inflammation after RF treatment of skin inflammatory response through TLRs/HMGB1, and RF treatment criteria are expected to provide good hints in clinical trials.

Reviewer 2 Report
This is interesting paper, which however requires revisions.
The information that studies were approved by local IRB (local ethics committee) should be listed in the proper section. I believe that informed consent was required since pictures of the patient are shown.
The introduction and discussion has to be improved.
Specifically the reader should be informed on diverse action of UV (Endocrinology 159(5), 1992-2007. 2018). Also note that UVB also induces immunosuppressive molecules including IL-10, POMC derived peptides and glucocorticoids (Amer J Physiol Endocrinol Metab 301:E484-E493, 2011; ; J Invest Dermatol, 135:1638-48, 2015; Photochem Photobiol. 93:1008–1015, 2017). These issues should be discussed.
I am surprised that authors discuss postinflammatory pigmenta alteration but do not mention or discuss UVB induced melanogenesis (Physiol Rev 84, 1155-1228, 2004).
English has to be improved.
Number of citations is low and not always representative for the field.
Author Response
Response to Reviewer 2 Comments
This is interesting paper, which however requires revisions.
Point 1: The information that studies were approved by local IRB (local ethics committee) should be listed in the proper section. I believe that informed consent was required since pictures of the patient are shown.
Response 1: We appreciated with this comment. We reported a case of 2 patients and consents has been obtained from all patients and example (top case 1; bottom case 2) can be found below (supporting figure 1).
|
Supporting figure 1 (case 1)
Supporting figure 1 (case 2)
|
Point 2: Specifically the reader should be informed on diverse action of UV (Endocrinology 159(5), 1992-2007. 2018). Also note that UVB also induces immunosuppressive molecules including IL-10, POMC derived peptides and glucocorticoids (Amer J Physiol Endocrinol Metab 301:E484-E493, 2011; ; J Invest Dermatol, 135:1638-48, 2015; Photochem Photobiol. 93:1008–1015, 2017). These issues should be discussed.
Response 2: We appreciated with kind comment. As you pointed out, we checked the reference papers you recommended and added the content in Introduction section.
|
Introduction section, Page 1, Line 43 in revised manuscript [Previous version] In UVB-exposed epidermis, epidermal growth factor receptor (EGFR) is increased to suppress apoptosis and leads keratinocyte proliferation which lead epidermal hyperplasia associated with increased cyclin D expression [6]. Interestingly, skin stress by UV exposure can trigger systemic neuroendocrine with its direct hormonal, neural and chemical consequences [7]. On the other hands, the nuclear factor kappa-light-chain-enhancer of activated B cells (NF-κB), a prototypical pro-inflammatory regulator is also activated, and it controls cell growth through activation of cyclin D1 expression [7]. [New version] In the UVB-exposed epidermis, epidermal growth factor receptor is increased to suppress apoptosis and to lead keratinocyte proliferation, which in turn leads to epidermal hyperplasia associated with increased cyclin D expression [7]. Interestingly, skin stress due to UVB exposure can trigger systemic neuroendocrine responses with direct hormonal, neural, and chemical consequences [8]. UVB exposure stimulates secretion of β-endorphins, corticosterone, urocortin, and adrenocorticotropic hormones through the hypothalamic-pituitary-adrenal axis, all of which are related to immunosuppressive effects [9,10]. In contrast, the nuclear factor kappa-light-chain-enhancer of activated B cells (NF-κB), a prototypical pro-inflammatory regulator, is also activated, and controls cell growth through activation of cyclin D1 expression [11]. |
|
|
|
Reference section Page 9, Line 357 in revised manuscript [New version] 1. Slominski, A.T.; Zmijewski, M.A.; Plonka, P.M.; Szaflarski, J.P.; Paus, R. How UV Light Touches the Brain and Endocrine System Through Skin, and Why. Endocrinology. 2018, 159, 1992-2007. 2. Skobowiat, C.; Slominski, A.T. UVB Activates Hypothalamic-Pituitary-Adrenal Axis in C57BL/6 Mice. J Invest Dermatol. 2015, 135, 1638-1648. 3. Skobowiat, C.; Postlethwaite, A.E.; Slominski, A.T. Skin Exposure to Ultraviolet B Rapidly Activates Systemic Neuroendocrine and Immunosuppressive Responses. Photochem Photobiol. 2017, 93, 1008-1015. |
Point 3: I am surprised that authors discuss postinflammatory pigmenta alteration but do not mention or discuss UVB induced melanogenesis (Physiol Rev 84, 1155-1228, 2004).
Response 3: We appreciated with kind comment. This study did not mention melanogenesis as it was a study on the reduction of inflammation by UVB, not on the results of the study on melanogenesis and inhibition by UVB. However, since melanogenesis is the most common change in skin caused by UVB, we briefly mentioned melanin formation. we checked the reference paper you recommended and added the content in introduction section.
|
Introduction section, Page 1, Line 35 in revised manuscript [Previous version] Skin absorbed ultraviolet-B (UVB, 290–320 nm) of the solar spectrum and might cause skin inflammation [1] and DNA mutations [2], suppress the immune system [3]. [New version] Skin absorption of ultraviolet B (UVB; 290–320 nm) can lead to skin inflammation [1], prompt DNA mutations [2], suppress the immune system [3], and cause melanogenesis [4]. When skin is exposed to UVB, melanocytes located in the stratum basale of the epidermis layer produce melanin in melanosomes, which is transferred from melanocytes to keratinocytes [4]. |
Point 4: English has to be improved.
Response 4: We appreciated with this comment. As you pointed out, the overall English revision was made through native speakers of whole manuscript and attached the proofread in English.
|
English certificate (see the attachment)
|
|
|
|
Whole manuscript
|
Point 5: Number of citations is low and not always representative for the field.
Response 5: We appreciated with scrupulous point. As your comment, 12 references were added in overall manuscript and added reference papers marked red color in References section.
|
References section, Page 9, Line 356 in revised manuscript [Previous version] 1. Clydesdale, G.J.; Dandie, G.W.; Muller, H.K. Ultraviolet light induced injury: immunological and inflammatory effects. Immunol Cell Biol 2001, 79, 547-568. 2. Melnikova, V.O.; Ananthaswamy, H.N. Cellular and molecular events leading to the development of skin cancer. Mutat Res 2005, 571, 91-106. 3. Fisher, M.S.; Kripke M.L. Suppressor T lymphocytes control the development of primary skin cancers in ultraviolet-irradiated mice. Science 1982, 216, 1133-1134. 4. Köck, A.; Schwarz, T.; Kirnbauer, R. et al. Human keratinocytes are a source for tumor necrosis factor alpha: evidence for synthesis and release upon stimulation with endotoxin or ultraviolet light. J Exp Med 1990, 172, 1609-1614. 5. El-Abaseri, T.B.; Putta, S.; Hansen, L.A. Ultraviolet irradiation induces keratinocyte proliferation and epidermal hyperplasia through the activation of the epidermal growth factor receptor. Carcinogenesis 2006, 27, 225-231. 6. Guttridge, D.C.; Albanese, C.; Reuther, J.Y.; Pestell, R.G.; Baldwin, A.S. Jr. NF-kappaB controls cell growth and differentiation through transcriptional regulation of cyclin D1. Mol Cell Biol 1999, 19, 5785-5799. 7. Freedberg, I.M.; Tomic-Canic, M.; Komine, M.; Blumenberg, M. Keratins and the keratinocyte activation cycle. J Invest Dermatol 2001, 116, 633-640. 8. Wu, L.; Chen, X.; Zhao, J. et al. A novel IL-17 signaling pathway controlling keratinocyte proliferation and tumorigenesis via the TRAF4-ERK5 axis. J Exp Med 2015, 212, 1571-1587. 9. Nygaard, U.; van den Bogaard, E.H.; Niehues H. et al. The "Alarmins" HMBG1 and IL-33 Downregulate Structural Skin Barrier Proteins and Impair Epidermal Growth. Acta Derm Venereol 2017, 97, 305-312 10. Erlandsson, Harris.H.; Andersson, U. Mini-review: The nuclear protein HMGB1 as a proinflammatory mediator. Eur J Immunol 2004, 34, 1503-1512. 11. Zhang, C.; Dong, H.; Chen, F.; Wang, Y.; Ma, J.; Wang, G. The HMGB1-RAGE/TLR-TNF-α signaling pathway may contribute to kidney injury induced by hypoxia. Exp Ther Med 2019, 17, 17-26. 12. Alexis, A.F.; Sergay, A.B.; Taylor, S.C. Common dermatologic disorders in skin of color: a comparative practice survey. Cutis 2007, 80, 387-394. 13. Masu, S.; Seiji, M. Pigmentary incontinence in fixed drug eruptions. Histologic and electron microscopic findings. J Am Acad Dermatol 1983, 8, 525-532. 14. Kim, D.; Lockey, R. Dermatology for the allergist. World Allergy Organ J 2010, 3, 202-215. 15. Kim, H.M.; Lee, M.J. Therapeutic Efficacy and Safety of Invasive Pulsed-Type Bipolar Alternating Current Radiofrequency on Melasma and Rebound Hyperpigmentation. Medical Lasers 2017, 6, 17-23. 16. Chung, J.Y.; Lee, J.H. Adverse Events after Noninvasive Radiofrequency Treatment for Cosmetic Uses. Medical Lasers 2015, 4, 16-19. 17. Sesto, A.; Navarro, M.; Burslem, F.; Jorcano, J.L. Analysis of the ultraviolet B response in primary human keratinocytes using oligonucleotide microarrays. Proc Natl Acad Sci U S A 2002, 99, 2965-2970. 18. Ryser, S.; Schuppli, M.; Gauthier, B. et al. UVB-induced skin inflammation and cutaneous tissue injury is dependent on the MHC class I-like protein, CD1d. J Invest Dermatol 2014, 134, 192-202. 19. Miller, L.S. Toll-like receptors in skin. Adv Dermatol 2008, 24, 71-87. 20. Lee, K.Y.; Kim, B.C.; Han, N.K. et al. Effects of combined radiofrequency radiation exposure on the cell cycle and its regulatory proteins. Bioelectromagnetics 2011, 32, 169-178. 21. Bertheloot, D.; Latz, E. HMGB1, IL-1α, IL-33 and S100 proteins: dual-function alarmins. Cell Mol Immunol 2017, 14, 43-64. 22. Chaowattanapanit, S.; Silpa-Archa, N.; Kohli, I.; Lim, H.W.; Hamzavi, I. Postinflammatory hyperpigmentation: A comprehensive overview: Treatment options and prevention. J Am Acad Dermatol 2017, 77, 607-621. 23. Calderhead, R.G. Photobiological Basics of Photomedicine: A Work of Art Still in Progress. Medical Lasers 2017, 6, 45-57. 24. Davis, E.C.; Callender, V.D. Postinflammatory hyperpigmentation: a review of the epidemiology, clinical features, and treatment options in skin of color. J Clin Aesthet Dermatol 2010, 3, 20-31 [New version] References 1. Clydesdale, G.J.; Dandie, G.W.; Muller, H.K. Ultraviolet light induced injury: immunological and inflammatory effects. Immunol Cell Biol 2001, 79, 547-568. 2. Melnikova, V.O.; Ananthaswamy, H.N. Cellular and molecular events leading to the development of skin cancer. Mutat Res 2005, 571, 91-106. 3. Fisher, M.S.; Kripke M.L. Suppressor T lymphocytes control the development of primary skin cancers in ultraviolet-irradiated mice. Science 1982, 216, 1133-1134. 4. Slominski, A.; Tobin, D.J.; Shibahara, S.; Wortsman, J. Melanin pigmentation in mammalian skin and its hormonal regulation. Physiol Rev. 2004, 84, 1155-228. 5. Köck, A.; Schwarz, T.; Kirnbauer, R. et al. Human keratinocytes are a source for tumor necrosis factor alpha: evidence for synthesis and release upon stimulation with endotoxin or ultraviolet light. J Exp Med 1990, 172, 1609-1614. 6. Luger, T.A.; Schwarz, T. Evidence for an Epidermal Cytokine Network. J. Invest. Dermatol. 1990, 95, 100S104S. 7. El-Abaseri, T.B.; Putta, S.; Hansen, L.A. Ultraviolet irradiation induces keratinocyte proliferation and epidermal hyperplasia through the activation of the epidermal growth factor receptor. Carcinogenesis 2006, 27, 225-231. 8. Slominski, A.T.; Zmijewski, M.A.; Plonka, P.M.; Szaflarski, J.P.; Paus, R. How UV Light Touches the Brain and Endocrine System Through Skin, and Why. Endocrinology. 2018, 159, 1992-2007. 9. Skobowiat, C.; Slominski, A.T. UVB Activates Hypothalamic-Pituitary-Adrenal Axis in C57BL/6 Mice. J Invest Dermatol. 2015, 135, 1638-1648. 10. Skobowiat, C.; Postlethwaite, A.E.; Slominski, A.T. Skin Exposure to Ultraviolet B Rapidly Activates Systemic Neuroendocrine and Immunosuppressive Responses. Photochem Photobiol. 2017, 93, 1008-1015. 11. Guttridge, D.C.; Albanese, C.; Reuther, J.Y.; Pestell, R.G.; Baldwin, A.S. Jr. NF-kappaB controls cell growth and differentiation through transcriptional regulation of cyclin D1. Mol Cell Biol 1999, 19, 5785-5799. 12. Freedberg, I.M.; Tomic-Canic, M.; Komine, M.; Blumenberg, M. Keratins and the keratinocyte activation cycle. J Invest Dermatol 2001, 116, 633-640. 13. Wu, L.; Chen, X.; Zhao, J. et al. A novel IL-17 signaling pathway controlling keratinocyte proliferation and tumorigenesis via the TRAF4-ERK5 axis. J Exp Med 2015, 212, 1571-1587. 14. Nygaard, U.; van den Bogaard, E.H.; Niehues H. et al. The "Alarmins" HMBG1 and IL-33 Downregulate Structural Skin Barrier Proteins and Impair Epidermal Growth. Acta Derm Venereol 2017, 97, 305-312 15. Erlandsson, Harris.H.; Andersson, U. Mini-review: The nuclear protein HMGB1 as a proinflammatory mediator. Eur J Immunol 2004, 34, 1503-1512. 16. Zhang, C.; Dong, H.; Chen, F.; Wang, Y.; Ma, J.; Wang, G. The HMGB1-RAGE/TLR-TNF-α signaling pathway may contribute to kidney injury induced by hypoxia. Exp Ther Med 2019, 17, 17-26. 17. Taticchi, A.; Urbani, S.; Albi, E. et al. In Vitro Anti-Inflammatory Effects of Phenolic Compounds from Moraiolo Virgin Olive Oil (MVOO) in Brain Cells via Regulating the TLR4/NLRP3 Axis. Molecules 2019, 24, 4523. 18. Alexis, A.F.; Sergay, A.B.; Taylor, S.C. Common dermatologic disorders in skin of color: a comparative practice survey. Cutis 2007, 80, 387-394. 19. Masu, S.; Seiji, M. Pigmentary incontinence in fixed drug eruptions. Histologic and electron microscopic findings. J Am Acad Dermatol 1983, 8, 525-532. 20. Taylor, S.; Grimes, P.; Lim, J.; Im. S.; Lui, H. Postinflammatory hyperpigmentation. J Cutan Med Surg 2009, 183191. 21. Kim, D.; Lockey, R. Dermatology for the allergist. World Allergy Organ J 2010, 3, 202-215. 22. Kim, H.M.; Lee, M.J. Therapeutic Efficacy and Safety of Invasive Pulsed-Type Bipolar Alternating Current Radiofrequency on Melasma and Rebound Hyperpigmentation. Medical Lasers 2017, 6, 17-23. 23. Cho, S.B.; Na. J.; Zheng, J.M. et al. In vivo skin reactions from pulsed-type, bipolar, alternating current radiofrequency treatment using invasive noninsulated electrodes. Skin Res Technol 2018, 318-325. 24. Chung, J.Y.; Lee, J.H. Adverse Events after Noninvasive Radiofrequency Treatment for Cosmetic Uses. Medical Lasers 2015, 4, 16-19. 25. Sesto, A.; Navarro, M.; Burslem, F.; Jorcano, J.L. Analysis of the ultraviolet B response in primary human keratinocytes using oligonucleotide microarrays. Proc Natl Acad Sci U S A 2002, 99, 2965-2970. 26. Yoshizumi, M.; Nakamura, T.; Kato, M. et al. Release of cytokines/chemokines and cell death in UVB-irradiated human keratinocytes, HaCaT. Cell Biol. Int. 2008, 32, 14051411. 27. Ryser, S.; Schuppli, M.; Gauthier, B. et al. UVB-induced skin inflammation and cutaneous tissue injury is dependent on the MHC class I-like protein, CD1d. J Invest Dermatol 2014, 134, 192-202. 28. Bashir, M.M.; Sharma, M.R.; Werth, V.P. UVB and proinflammatory cytokines synergistically activate TNF-alpha production in keratinocytes through enhanced gene transcription. J. Invest. Dermatol. 2009, 9941001. 29. Miller, L.S. Toll-like receptors in skin. Adv Dermatol 2008, 24, 71-87. 30. Seok, J.; Kim, J.H.; Kim, J.M. et al. Effects of Intradermal Radiofrequency Treatment and Intense Pulsed Light Therapy in an Acne-induced Rabbit Ear Model. Sci. Rep. 2019, 9, 5056. 31. Michalczyk, T.; Biedermann, T.; Bottcher-Haberzeth, S.; Klar, A.S.; Meuli, M.; Reichmann, E. UVB exposure of a humanized skin model reveals unexpected dynamic of keratinocyte proliferation and Wnt inhibitor balancing. J. Tissue Eng. Regen. Med. 2018, 12, 505515. 32. Lee, K.Y.; Kim, B.C.; Han, N.K. et al. Effects of combined radiofrequency radiation exposure on the cell cycle and its regulatory proteins. Bioelectromagnetics 2011, 32, 169-178. 33. Bertheloot, D.; Latz, E. HMGB1, IL-1α, IL-33 and S100 proteins: dual-function alarmins. Cell Mol Immunol 2017, 14, 43-64. 34. Chaowattanapanit, S.; Silpa-Archa, N.; Kohli, I.; Lim, H.W.; Hamzavi, I. Postinflammatory hyperpigmentation: A comprehensive overview: Treatment options and prevention. J Am Acad Dermatol 2017, 77, 607-621. 35. Calderhead, R.G. Photobiological Basics of Photomedicine: A Work of Art Still in Progress. Medical Lasers 2017, 6, 45-57. 36. Davis, E.C.; Callender, V.D. Postinflammatory hyperpigmentation: a review of the epidemiology, clinical features, and treatment options in skin of color. J Clin Aesthet Dermatol 2010, 3, 20-31 |

Round 2
Reviewer 2 Report
The authors adequately replied to the critique